Development and external validation of a multivariate model for predicting pneumonia in patients receiving maintenance hemodialysis: a retrospective study

Yang Xiao Hua 1
Zhang Ju 1
Xie Xi Sheng xishengxie2023@163.com 2
Tang Wen Wu tww-2-17@163.com 1
1 Department of Nephrology, Guangyuan Central Hospital , Guangyuan City , China
2 Department of Nephrology, Nanchong Central Hospital Affiliated to North Sichuan Medical College , Nanchong City , China
Upadhyay Rohit
Electronic publication date: 2025 Oct 9
Publication date: 2025
Volume: 13
Electronic Location ID: e20070
Received 2025 Feb 26; Accepted 2025 Aug 21
Copyright: ©2025 Yang et al.
Copyright year: 2025
Copyright holder: Yang et al.
License: This is an open access article distributed under the terms of the Creative Commons Attribution License, which permits unrestricted use, distribution, reproduction and adaptation in any medium and for any purpose provided that it is properly attributed. For attribution, the original author(s), title, publication source (PeerJ) and either DOI or URL of the article must be cited.
License URL: https://creativecommons.org/licenses/by/4.0/

Keywords: Patients on MHD, Pneumonia, Predictive model, External validation, Retrospective study

Funding: The Sichuan Provincial Administration of Traditional Chinese Medicine Research Fund 2020JC0079 Sichuan Provincial Department of Science and Technology Research Special Fund 2021YFS0259 Nanchong Science and Technology Plan Project 22JCYJPT0005 22JCYJPT0011 This research was supported by the Sichuan Provincial Administration of Traditional Chinese Medicine Research Fund (2020JC0079); Sichuan Provincial Department of Science and Technology Research Special Fund (2021YFS0259); Nanchong Science and Technology Plan Project (22JCYJPT0005); Nanchong Science and Technology Plan Project (22JCYJPT0011). The funders had no role in study design, data collection and analysis, decision to publish, or preparation of the manuscript.

==============================
Background

Patients receiving maintenance hemodialysis (MHD) who develop pneumonia experience substantially elevated risks of hospitalization and mortality, while also incurring significantly heightened healthcare-related financial burdens. Our goal is to establish a forecasting model to assess the individual risk of pneumonia in patients undergoing MHD.

Materials and Methods

A retrospective analysis was carried out between January 2018 and November 2024, involving 405 MHD patients from two medical centers. The variables underwent adjustment through multivariate Cox regression analysis, and the forecasting model was created and verified.

Results

The median follow-up time of the external validation set was 35 months (interquartile range: 20–43), and the median follow-up time of the modeling set was 22 months (12–24). The event happened in 101 (34.83%) out of 290 patients in the modeling set and 45 (39.13%) out of 115 patients in the external validation set. The model predictors included history of diabetes and coronary heart disease; serous effusion; white blood cell; albumin-globulin ratio; left ventricular mass index, and age. The C-index was 0.753 (0.684, 0.822) for the external validation set and 0.772 (95% CI [0.724–0.821]) for the modeling set. The model showed excellent calibration ability throughout the risk spectrum, and decision curve analysis showed that it could maximize the prognosis of patients.

Conclusion

The created predictive model provided a precise, individualized evaluation of pneumonia risk in patients with MHD. It can be used to identify individuals at high risk of pulmonary infection in patients undergoing MHD and guide their treatment and prognosis follow-up.

Introduction

According to statistics, infections are one of the leading causes of hospitalization for patients with maintenance hemodialysis (MHD), second only to cardiovascular disease (Gandra et al., 2021; Guo et al., 2008). Among infections, pneumonia ranks first, with a high incidence rate of 28% within the first year of dialysis treatment (Gandra et al., 2021; Guo et al., 2008). According to statistics, the cumulative probability of pneumonia hospitalization within one year for MHD patients is 9%, and it increases to 36% within five years (Slinin, Foley & Collins, 2006). The one-year and five-year mortality rates for patients with MHD after contracting pneumonia are 45% and 83%, respectively (Slinin, Foley & Collins, 2006).

In current clinical treatment, MHD treatment is the primary method for eliminating metabolites from the blood and stabilizing the internal environment of dialysis patients to extend their survival, despite causing nutrient and trace element loss, immune dysfunction, weakening barrier function against pathogens, and heightened infection risk (Di Pasquale et al., 2019; Dou et al., 2022; Pettigrew, Tanner & Harris, 2021; Toma, Naka & Iseki, 2021; Vanholder & Ringoir, 1993). In addition, dialysis fluid and dialysis cuff catheters used during dialysis treatment, or puncture damage to arteriovenous fistulas, can also be a potential route of bacterial infection (Machiba et al., 2022; Tavakoli et al., 2021; Toma, Naka & Iseki, 2021). In addition, there are many complications such as heart failure (HF), diabetes and coronary heart disease in patients with MHD, which are difficult to control, and also increase the challenge for the prevention of pulmonary infection (Torres et al., 2021). Therefore, early identification of high-risk MHD patients with pulmonary infection, accurate and personalized dialysis and drug treatment plan design is essential.

Forecasting pulmonary infections in MHD patients using existing models poses a challenge. First of all, the previously published predictive models are mainly developed for the general population or chronic kidney disease (CKD) population. They only include routine demographic and laboratory examination characteristics, and do not consider the important characteristics of dialysis treatment (dialysis mode, dialysis frequency, dialysis duration), heart failure (HF) management (cardiac biomarkers, echocardiography) and prognosis in patients with MHD (Deng et al., 2024; Gearhart et al., 2019; Markussen et al., 2024; Shirata et al., 2021). Secondly, the above studies focused more on the prognosis of patients with pneumonia (death or rehospitalization), rather than the early prediction of pulmonary infection (Deng et al., 2024; Gearhart et al., 2019; Markussen et al., 2024; Shirata et al., 2021). Furthermore, the above prediction models were not verified in the independent cohort (Deng et al., 2024; Gearhart et al., 2019; Markussen et al., 2024; Shirata et al., 2021). Pulmonary infection is the primary infectious disease among patients undergoing MHD and a significant prognostic factor; however, the relevant risk factors for pulmonary infections in these patients remain unclear (Torres et al., 2021).

The purpose of this study was to establish and validate a predictive model for pneumonia in maintenance hemodialysis patients to achieve accurate assessment of individual pneumonia risk. The aim is to improve early detection and intervention of pneumonia by healthcare providers, ultimately enhancing the prognosis of patients receiving MHD.

Materials & Methods

Materials

Subjects

This retrospective analysis examined data from 405 patients undergoing MHD at Guangyuan Central Hospital and Nanchong Central Hospital, between January 2018 and November 2024 (Fig. 1).

Figure 1 Research flow chart.

Inclusion criteria

(1) The age should be over 18 years old; (2) Patients diagnosed with CKD5 receiving dialysis treatment, fulfilling the requirements established by the Kidney Disease Improving Global Outcomes guidelines. In addition, regular hemodialysis must be performed for at least 3 months, frequency ≥ 2 times (weekly); (3) Patients who initially began to receive MHD treatment.

Exclusion criteria

(i) Patients were considered ineligible if they had chronic obstructive pulmonary disease, severe bronchiectasis, a history of malignancy, hepatic conditions, cystic fibrosis, recent hospitalization within the 14 days preceding admission, or were under a palliative approach (i.e., with a life expectancy of less than 2 weeks); (ii) who had received a kidney transplant; (iii) who refused to participate, could not cooperate or had incomplete clinical data.

Methods

Diagnostic criteria for pneumonia

Met at least three of the following criteria: ① Recent sputum production, cough, or exacerbation of existing respiratory diseases, with or without purulent sputum, dyspnea, chest pain, or hemoptysis; ② fever; ③ signs of lung consolidation and/or audible moist rales; ④ peripheral white blood cell count<4 × 109/L or >10 × 109/L, with or without a left shift in the nucleus; ⑤ chest imaging revealed new patchy infiltration, lobar/segmental consolidation, ground-glass opacities, or interstitial changes, with or without pleural effusion; ⑥ excluding chronic obstructive pulmonary disease, pulmonary tuberculosis, lung tumors, non-infectious non-interstitial diseases, pulmonary edema, atelectasis, pulmonary embolism, pulmonary eosinophilic infiltration, pulmonary vasculitis, and positive throat swab results for coronavirus disease 2019 (Markussen et al., 2024; Serigstad et al., 2022; White et al., 2024). This study follows the Declaration of Helsinki and has been approved by the Ethics Committee of Guangyuan Central Hospital (No: 2024-08 and date of approval 11.19.2024). However, written consent was not necessary given the retrospective nature of the study.

Starting point and endpoint of observation

We plan that each patient starts from the beginning of MHD treatment as the starting point of observation. This setting ensures that the prediction results of our model are in the early stage of the disease rather than the end stage of the disease. Before the patient’s first MHD treatment, each center will conduct a comprehensive and detailed assessment of them to prepare for subsequent treatment (including relevant laboratory tests and ultrasound examinations), rather than based on the patient’s disease status. The observation endpoint was considered reached in the presence of pneumonia or upon completion of the follow-up period. We defined the follow-up period as a period of time between the starting point of observation and the end point of observation, lasting at least 3 months. Pneumonia was the primary outcome of our study. A patient with MHD must regularly go to each center for dialysis treatment. Once the treatment is interrupted, the staff of each center will immediately contact the patient or his family by telephone, and record the cause of the patient’s treatment interruption in detail in the system. It is the particularity of the treatment of patients with MHD mentioned above that ensures that there are almost no lost follow-up events in our study cohort.

Study indicators

This study identified candidate predictors based on clinical guidelines and literature review. Patient information, details of dialysis treatment, and clinically relevant data were retrieved from the electronic medical record systems of each center. We collected the variables of patients receiving MHD for the first time. Our study contains a total of 42 candidate predictors, which encompassed various aspects such as (1) general and dialysis-related information. These included factors such as sex, age, smoking or drinking habits, type of dialysis vascular access, frequency (weekly), duration (hours per session) of dialysis, urea reduction ratio (URR), single-pool Kt/V (spKt/V; K, urea dialytic clearance; t, dialysis time; V, urea distribution volume), and ultrafiltration rate (UFR). (2) Clinical data: body mass index (BMI), blood pressure, Basic diseases leading to renal failure (diabetes, nephritis, hypertension, lupus nephritis, polycystic kidney and so on), history of related previous diseases (coronary heart disease (CHD), cerebral apoplexy, hypertension, diabetes, history of fracture surgery), serous cavity effusion, pulmonary artery hypertension. (3) Laboratory examination: white blood cell (WBC), neutrophil ratio (NEU%), lymphocyte ratio (LYM%), hemoglobin (HGB), C-reactive protein (CRP), albumin-globulin ratio (A/G), serum calcium (Ca), serum phosphorus (P), TC (total cholesterol), serum creatinine (Scr), parathyroid hormone (PTH), platelet to high-density lipoprotein cholesterol ratio (PHR), N-terminal prohormone of brain natriuretic peptide (NT-proBNP), venous blood glucose. (4) Echocardiographic data: left ventricular ejection fraction (LVEF), pulmonary artery hypertension, and left ventricular mass index (LVMI, calculated according to the formula of American Society of Echocardiography (Lang et al., 2015)). (5) Medication: antihypertensive drugs (angiotension converting enzyme inhibitor/ angiotensin II receptor blocker/ calcium antagonists/α-receptor blocker, antidiabetic drugs (oral hypoglyceimic agents/ insulin, calcium supplements drugs (calcium tablets/vitamin). Due to the obvious multicollinearity between ‘antihypertensive drugs’ and ‘baseline blood pressure’, ‘calcium, vitamin D’ and ‘parathyroid hormone, blood calcium’, ‘insulin or hypoglycemic drugs’ and ‘Basic diseases leading to renal failure (especially diabetes)’ variables; secondly, due to the single factor Cox analysis, the correlation between it and the outcome events was not significant (P > 0.05); Regarding the dialysis-related predictor, guidelines typically recommend single-pool Kt/V (spKt/V; K, urea dialytic clearance; t, dialysis time; V, urea distribution volume) as the preferred indicator of dialysis adequacy. However, recent studies have shown that spKt/V and URR have similar prognostic values for all-cause mortality (Chen et al., 2023). We performed correlation and Cox univariate analysis on spKt/V and URR and found significant collinearity between the two measures. However, the association between spKt/V and outcome events was insignificant (P = 0.423). According to the expert clinical consensus, the most informative and representative variables are selected from the highly relevant variables. Therefore, we chose to exclude six variables, such as basic diseases leading to renal failure, drug-related variables (antihypertensive drugs/antidiabetic drugs/calcium supplements drugs), and intravenous blood glucose, and finally only 36 variables were retained for subsequent analysis. The results of serous effusion were derived from pericardial and pleural effusion in cardiac ultrasound and chest CT examination. The HCO3−<22 mmol/L in blood gas analysis was defined as metabolic acidosis (Kraut & Madias, 2018). Venous blood samples were collected after overnight fasting for ≥8 h before MHD treatment. The laboratory and imaging center of each center completed the corresponding laboratory examination or echocardiography. Patients in both centers were dialyzed using GambroAK9, Gambro PA14016, Fresenius5008 s, and WEGO (DBB-06S) dialyzer.

Construction and evaluation of predictive models

From all the independent variables, characteristic factors were selected. Patients receiving maintenance hemodialysis (MHD) were separated into modeling sets (n = 290) at Guangyuan Central Hospital and external validation sets (n = 115) at Nanchong Central Hospital. The significance of each index was assessed and compared across different models in both groups. Following a series of detailed steps, the best model was subjected to further evaluation and validation: (1) Identification of characteristic factors through screening: Least Absolute Shrinkage and Selection Operator (Lasso) regression analysis was performed using the Glmnet package (R software, version 4.1.2), and multivariate Cox regression analysis was performed using SPSS (version 26.0, IBM, USA). P < 0.05 was considered significant. (2) Pearson correlation coefficient (r) was calculated to evaluate the correlation between predictors. When r > 0.7, it was considered that there was multicollinearity between variables (Python, Sklearn 0.22.1) (Sauerbrei, Royston & Binder, 2007) (3) For model visualization, the nomogram was created using the ‘logreg6.2.0’ R software package, and the forest plot was generated using the ‘ggplot2’ package. (4) The ‘survivalROC’ package (R software, version 4.2.2) was used to draw the receiver operating characteristic (ROC) curve and evaluate the accuracy of the model. The area under the curve and C statistics of the modeling set and the external validation set are calculated respectively (Eom et al., 2015). To measure the predictive ability of the model and assess the concordance between predicted and actual risks, calibration curves were drawn using the ‘rms’ (R-software, version 6.2.0) and ‘timeROC’ packages (R-software, version 0.4) (Eom et al., 2015). Decision curve analysis (DCA) was performed using the ‘ggDCA ’ package (R software) (Vickers & Elkin, 2006), and Kaplan–Meier (K–M) curves were generated using the ‘survminer’ package (R software) to comprehensively evaluate the clinical value of our model.

Statistical analysis

A comparison of variables was performed between the modeling and external validation sets. The distribution of continuous variables was represented by median and interquartile range (IQR), and continuous variables were compared using the Mann–Whitney U test. The distribution of categorical variables was expressed by count and percentage, and the comparison between groups was performed by chi-square test. Two-sided test P < 0.05 was considered significant. Statistical analyses were conducted using SPSS version 26.0 (IBM Corp., Armonk, NY, USA) and the R-software package version 4.2.2. The proportional hazards assumption was assessed using Schoenfeld residuals (Harrell, 2015). When cleaning the data, it was found that there were 9.63% −10.62 defects in TC, TG, LVEF, LVMI and NT-proBNP, and the defect variables were filled by multiple imputation (All statistical analyses were conducted using R version 4.2.3 and Python version 3.11.4), and compare the significant impact on data distribution before and after data filling (Fig. S1).

Results

Demographic and clinical characteristics of study population

According to the inclusion and exclusion criteria, 405 patients with MHD were included in our study. Table 1 summarizes the distribution of baseline data for the modeling set and the external validation set. There were significant differences in smoking or drinking, hemodialysis vascular access, hypertension, metabolic acidosis, pulmonary artery hypertension, serous effusion, URR, WBC, NEU%, LYM%, A/G, scr, P, ln (NT-proBNP) between the two groups (P < 0.05). The median follow-up time of modeling set and external validation set was 22 months (IQR 12-45) and 35 months (20-43), respectively. The incidence of pneumonia in the modeling set was 34.83% (101/290), and the incidence of pneumonia in the external validation set was 39.13% (45/105). There was no loss of follow-up or withdrawal from the study.

Table 1 Baseline characteristics of the modeling and external validation set.

Variables	Modeling sets
(n = 290)	External validation sets
(n = 115)	Z	P	
Sex, n (%)					
Female	108 (37.241)	47 (40.870)	0.459	0.498	
Male	182 (62.759)	68 (59.130)			
Smoking or Drinking, n (%)					
No	219 (75.517)	111 (96.522)	24.077	<0.001	
Yes	71 (24.483)	4 (3.478)			
Hemodialysis vascular access, n (%)					
Autogenous arteriovenous fistula	175 (60.345)	113 (98.261)	57.626	<0.001	
Long-term cuff catheter	115 (39.655)	2 (1.739)			
Dialysis frequency (weekly), n (%)					
<3	54 (18.621)	11 (9.565)	5.012	0.025	
≥3	236 (81.379)	104 (90.435)			
Duration of dialysis (h/time), n (%)					
3	21 (7.241)	13 (11.304)	2.199	0.333	
3.5	12 (4.138)	3 (2.609)			
4	257 (88.621)	99 (86.087)			
Reasons for entering dialysis, n (%)					
Diabetes	109 (37.586)	49 (42.609)	44.571	<0.001	
Nephritis	67 (23.103)	55 (47.826)			
Hypertension	13 (4.483)	5 (4.348)			
Others (Lupus nephritis, Polycystic kidney. etc.)	101 (34.828)	6(5.217)			
Cerebral apoplexy, n (%)					
No	250 (86.207)	101 (87.826)	0.187	0.666	
Yes	40 (13.793)	14 (12.174)			
Hypertension, n (%)					
No	104 (35.862)	9 (7.826)	37.83	<0.001	
Grade 1∼2	32 (11.034)	30 (26.087)			
Grade 3	154 (53.103)	76 (66.087)			
Diabetes, n (%)					
No	160 (55.172)	68 (59.130)	0.524	0.469	
Yes	130 (44.828)	47 (40.870)			
CHD, n (%)					
No	250 (86.207)	90 (78.261)	3.859	0.049	
Yes	40 (13.793)	25 (21.739)			
NYHA, n (%)					
0∼2	129 (44.483)	42 (36.522)	2.139	0.144	
3∼4	161 (55.517)	73 (63.478)			
Metabolic acidosis, n (%)					
No	254 (87.586)	62 (53.913)	54.455	<0.001	
Yes	36 (12.414)	53 (46.087)			
Pulmonary artery hypertension, n (%)					
No	238 (82.069)	33 (28.696)	105.956	<0.001	
Yes	52 (17.931)	82 (71.304)			
Serous effusio, n (%)					
No	222 (76.552)	56 (48.696)	29.685	<0.001	
Yes	68 (23.448)	59 (51.304)			
History of fracture surgery, n (%)					
No	286 (98.621)	109 (94.783)	5.037	0.025	
Yes	4 (1.379)	6 (5.217)			
Antihypertensive drugs, n (%)					
No	100 (34.483)	11 (9.565)	25.698	<0.001	
Yes	190 (65.517)	104 (90.435)			
Antidiabetic drugs, n (%)					
No	181 (62.414)	66 (57.391)	0.873	0.35	
Yes	109 (37.586)	49 (42.609)			
Calcium tablets or Vitamin D, n (%)					
No	20 (6.897)	4 (3.478)	1.726	0.189	
Yes	270 (93.103)	111 (96.522)			
Age (years), median [IQR]	57.000 [46.000, 68.000]	58.000 [46.000, 71.000]	−0.908	0.364	
BMI, median [IQR]	22.137 [20.386, 24.961]	23.280 [20.310, 25.160]	−1.343	0.179	
Pulse (bpm), median [IQR]	84.000 [76.000, 95.000]	82.000 [76.000, 90.000]	0.894	0.371	
SBP (mmHg), median [IQR]	154.000 [136.000, 170.000]	152.000 [137.000, 166.000]	0.867	0.386	
DBP (mmHg), median [IQR]	86.000 [77.000, 96.000]	82.000 [76.000, 92.000]	1.242	0.214	
URR (%), median [IQR]	68.190 [65.450, 71.480]	65.470 [63.850, 67.220]	6.466	<0.001	
Single-pool Kt/V [IQR]	1.430 [1.310, 1.520]	1.300[1.250, 1.330]	7.984	<0.001	
UFR (ml/kg*h), median [IQR]	9.797 [7.285, 12.460]	10.336 [8.750, 12.037]	−1.304	0.192	
WBC (109/L), median [IQR]	6.050 [4.720, 7.730]	7.160 [5.830, 8.980]	−4.25	<0.001	
NEU %, median [IQR]	74.600 [67.100, 80.800]	78.700 [74.200, 84.000]	−4.073	<0.001	
LYM %, median [IQR]	15.000 [10.300, 21.900]	11.400 [8.100, 15.600]	4.677	<0.001	
HGB (g/L), median [IQR]	83.000 [73.000, 96.000]	90.000 [78.000, 108.000]	−2.957	0.003	
CRP (mg/L), median [IQR]	4.610 [1.510, 15.800]	4.800 [1.600, 16.660]	0.164	0.87	
A/G, median [IQR]	1.470 [1.236, 1.650]	1.214 [1.024, 1.416]	6.93	<0.001	
Scr (μmol/L), median [IQR]	597.600 [435.000, 830.000]	806.000 [567.000, 1063.200]	−4.991	<0.001	
Ca (mmol/L), median [IQR]	1.950 [1.810, 2.110]	1.960 [1.730, 2.110]	0.541	0.589	
P (mmol/L), median [IQR]	1.580 [1.220, 2.080]	1.990 [1.390, 2.600]	−3.659	<0.001	
PTH (pmol/L), median [IQR]	259.410 [138.700, 378.880]	249.300 [140.200, 373.000]	0.243	0.808	
TC (mmol/L), median [IQR]	3.984 [3.400, 4.640]	4.130 [3.380, 4.960]	−1.116	0.265	
PHR (109 mmol/L2), median [IQR]	120.000 [87.912, 169.506]	139.423 [102.190, 210.870]	−2.538	0.011	
LVMI (g/m2), median [IQR]	119.826 [103.586, 139.631]	117.671 [99.794, 140.039]	0.963	0.336	
ln (NT-proBNP) (pg/mL), median [IQR]	8.629 [7.559, 9.686]	9.643 [8.423, 10.463]	−5.059	<0.001	
LVEF (%), median [IQR]	64.000 [58.000, 67.000]	65.000 [60.000, 70.000]	−2.366	0.018	
Notes.

CHD Coronary-heart-disease

BMI body mass index

SBP systolic blood pressure

DBP diastolic blood pressure

URR urea reduction ratio

spKt/V single-pool Kt/V

K urea dialytic clearance

t dialysis time

V urea distribution volume

UFR ultrafiltration rate

WBC white blood cell

NEU% neutrophil ratio

LYM% lymphocyte ratio

HGB hemoglobin

CRP C-reactive protein

A/G albumin-globulin ratio

Scr Serum creatinine

Ca serum calcium

P serum phosphorus

PTH parathyroid hormone

TC total cholesterol

PHR Platelet to high-density lipoprotein cholesterol ratio

LVMI left atrial diameter

NT-proBNP N-terminal prohormone of brain natriuretic peptide

LVEF left ventricular ejection fraction

Model construction

The correlation analysis of the screened predictors did not show any clear signs of collinearity (Fig. S2). The relationship between candidate variables and pneumonia outcome was preliminarily evaluated by Cox univariate regression analysis (Table S3). Through single factor regression analysis, 21 variables were statistically significant (P < 0.05). Lasso regression analysis with pulmonary infection as the endpoint reduced the number of predictors from 21 to 8 (Fig. 2).

Subsequently, multivariate Cox stepwise regression analysis was performed on the above eight variables to further adjust the confounding factors. Finally, seven variables were determined as predictors of the model (P < 0.05): Diabetes, CHD, serous effusion, age, WBC, A/G, LVMI (Fig. 3). The final Schoenfeld residual test confirms that the above variables meet the proportional risk hypothesis (Fig. S3).

We created a scoring system to comprehensively evaluate the predictive performance of the model. The system scores and sums each variable, and finally calculates the overall score. The point at which the line intersects the probability axis represents the probability of pneumonia at 12, 24, and 36 months (Fig. 4).

Multi-model comparison

The seven assessed variables were prioritized through AdaBoost regression, random forest regression, and Lasso regression analysis. The findings suggested that “CHD” was the most crucial predictive factor (Fig. 5). In order to evaluate the improvement of the model, Model 1 (constructed using six variables except ‘CHD’) and Model 2 (including seven variables) were compared.

Compared with model 1, the 1-year and 2-year DCA curves of model 2 in the modeling set and the external validation set show greater net benefits (Fig. 6). This suggests that the incorporation of “CHD” notably enhanced the model. Therefore, we believe that Model 2 is the best model and further verification is carried out.

Figure 2 LASSO regression.

Figure 3 Multivariate Cox regression analysis and forest plot.

Figure 4 Nomogram.

Figure 5 Ranking of predictor variable importance.

(A) LASSO regression; (B) Logistic regression; (C) XGBoost regression. CHD, Coronary-heart-disease; WBC, white blood cell; A/G, albumin-globulin ratio; LVMI, left ventricular mass index.

Figure 6 1-year and 2-year DCA curves.

(A) Modeling set; (B) External validation set. Clinical decision curve analysis (DCA) of the nomogram: the Y-axis represents fraction survival, the X-axis represents the threshold probability, the red line represents the net benefit of model 1, and the green line represents the net benefit of model 2.

The best model verification

The ROC curves of the modeling set and the external validation set were drawn, and the area under the curve of Model 2 was calculated. Over time, the time-AUC curve showed the stability of the prediction model, and the AUC values were maintained at about 0.8 (Fig. 7).

Figure 7 Time-ROC curve of the complete model.

(A–B) Modeling set; (C–D) External validation set.

The calibration curve closely follows the calibration chart of the reference line, suggesting the accurate prediction of the model in the whole risk range (Fig. 8).

Figure 8 Recalibration plots.

(A) 12 months. (B) 24 months. (C) 36 months.

Kaplan–Meier analysis was used to further verify the effect of risk factors on the occurrence of pneumonia in patients undergoing MHD (Fig. S4). WBC, A/G, LVMI, and Age were categorized according to cut-off values determined through ROC curve analysis.

Discussion

In this retrospective analysis, we created and validated a nomogram prediction model specifically for the risk of pneumonia in patients undergoing MHD. Based on our limited knowledge, this seems to be the first externally validated predictive model for pneumonia in MHD patients. Our results show that the model provides accurate and personalized risk assessment for individuals and has strong clinical practical value.

Compared with general patients, dialysis patients face more complex risk factors for pneumonia. These factors include rapid changes in hemodynamics and electrolyte levels, along with inadequate dialysis treatment. Therefore, prioritizing the identification of individuals at an increased risk of pneumonia-related hospitalization or mortality is essential. Numerous current prediction models do not consider patients undergoing MHD, making them unsuitable for this specific population (Ramirez et al., 2020; Toma, Naka & Iseki, 2021; Vanholder & Ringoir, 1992).

In line with prior research, our model took into account age, WBC, and albumin levels (Deng et al., 2024; Huang et al., 2024; Shirata et al., 2021). Moreover, our model integrated essential variables relevant to current MHD treatment management, including LVEF, LVMI, and URR assessment (Deng et al., 2024; Gearhart et al., 2019; Markussen et al., 2024; Shirata et al., 2021). Kaplan–Meier analysis showed that age > 60.5 years, LVMI ≥ 138.4 g/m2, WBC ≥ 6.71 × 109/L, A/G < 1.27, history of diabetes, and CHD were recognized as independent risk factors that elevate the likelihood of pneumonia.

Our results showed an 80% rise in pneumonia risk in patients with CHD and a 64.4% increase in pneumonia risk in patients with diabetes. There seems to be a two-way relationship between pneumonia and CHD. On the one hand, coronary artery disease increases the risk of pneumonia hospitalization (Kim et al., 2021); In contrast, pneumonia might also elevate the risk of acute coronary syndrome, such as myocardial infarction or unstable angina (Corrales-Medina et al., 2013). While pulmonary infections are typically viewed as acute occurrences, there is evidence suggesting that pneumonia is linked to cardiovascular complications that may manifest years later, notably acute coronary events such as left ventricular dysfunction, arrhythmia, ischemia, and infarction, as well as HF (Corrales-Medina et al., 2013). Regarding diabetes, a number of studies have shown that it increases the susceptibility of patients to pulmonary infection, and the infection rate will increase with age, poor blood glucose control, or deterioration of immune function (Critchley et al., 2018; Fazeli Farsani et al., 2015; Visca et al., 2018). The main pathogenesis of diabetes is closely related to immune dysfunction, including chemotaxis, phagocytosis, cytokine release and so on (Erener, 2020). The latest data show that, on the one hand, hypoproteinemia caused by diabetes and hemodialysis can affect the synthesis of immune factors and increase the risk of pulmonary infection (Oliver et al., 2022; Wand et al., 2022); on the other hand, the level of opportunistic Enterobacteriaceae in diabetic patients is high, and the intestinal barrier function is decreased, resulting in an increased risk of pathogen transmission (Anhê et al., 2020; Thaiss et al., 2018). Regarding serous effusion, current studies suggest that it may be the result of local inflammation, volume overload, and metabolic abnormalities caused by uremic toxin accumulation or infection (Ito & Akamatsu, 2024).

Among the included basic management indicators of cardiovascular and dialysis treatment, we found that LVMI had a high predictive value (HR: 1.007, 95% CI [1.002–1.019], P = 0.004). In the field of echocardiography, LVMI is often used to assess the health status of cardiac structure (left ventricular size and weight) and function; the increase of LVMI is closely related to heart diseases such as hypertension, coronary heart disease, left ventricular hypertrophy and HF (Heidenreich et al., 2022). The incidence of pneumonia in patients with HF is very high, which is about 2–3 times the expected incidence (Jobs et al., 2018; Shen et al., 2021). Water and sodium retention and toxic accumulation in patients with MHD increase cardiopulmonary function load and tissue structure damage. The decrease of cardiac pumping function in patients with HF is easy to cause pulmonary circulation disorder, which leads to pulmonary congestion and edema, pulmonary dysfunction and local defense function decline, resulting in pulmonary infection (Bartlett et al., 2019); Pulmonary infection can lead to the release of inflammatory factors and respiratory dysfunction in the body. At the same time, myocardial ischemia and hypoxia are aggravated, cardiac load is increased, and HF is aggravated (Mancini & Gibson, 2021). Current research and guidelines emphasize that in the secondary prevention of pulmonary infection, the most critical thing is to ensure that all heart (e.g., HF) and lung (e.g., chronic obstructive pulmonary disease, asthma) complications are treated in accordance with the guidelines, and to restart goal-directed therapy before discharge (Tralhão & Póvoa, 2020; Vaughn et al., 2024).

Standard laboratory tests contributed valuable prognostic information in our models. We found that higher levels of WBC and lower levels of A/G were closely related to the occurrence of pulmonary infection. The results of K-M curve analysis showed that ‘WBC’ with a cut-off value of 6.71 × 109/L and ‘A/G’ with a cut-off value of 1.27 could effectively distinguish the high-risk population of pulmonary infection. In clinical practice, WBC is often used as one of the diagnostic indicators of pulmonary inflammation (WBC 10 × 109/L), while A/G is used as a prognostic indicator of prostate cancer, lymphoma, and rectal cancer, with little attention paid to its prognostic value for pulmonary infection in MHD patients (An et al., 2022; Salciccia et al., 2022). Compared with the previous diagnostic criteria >10,000/µL, the results of this study showed that 6.71 as a cutoff seemed to effectively reduce the missed diagnosis of high-risk MHD patients with pulmonary infection. About A/G, albumin can reflect the nutritional status of the body, globulin can reflect the immune and inflammatory state, their proportion can be divided by albumin in serum total protein minus albumin value to evaluate (Soeters, Wolfe & Shenkin, 2019). The possible reasons for the decrease of the ratio are the decrease of albumin and the abnormal increase of globulin, which indicate the decrease of nutritional level, the hyperfunction of inflammatory state and the chronic inflammatory state (Kadatane et al., 2023; Salciccia et al., 2022). Inflammation in MHD is caused by a variety of mechanisms, including accumulation of pro-inflammatory cytokines due to clearance defects, uremia, oxidative stress, infection, volume overload, and dialysis treatment measures (Kadatane et al., 2023; Zoccali et al., 2017). Long-term dialysis patients have serious loss of nutrients, poor nutritional status, and severe reduction of immune function, so they are more prone to pulmonary infection (Wand et al., 2021). Therefore, the dynamic monitoring of WBC, A/G in patients with MHD should be strengthened. According to the 2019 Global Burden of Disease study, older adults, specifically those over the age of 70, are the demographic most impacted by pneumonia (2020). Our results also showed that age was an independent risk factor for pneumonia in patients with MHD, which was positively correlated with pulmonary infection. The risk of pneumonia increased by 28.8% for every 10-year increase in age (HR: 1.288, 95% CI [1.103–1.502], P = 0.001). However, our study found that MHD patients aged ≥ 60.5 years were at high risk of pneumonia by K-M curve analysis, which was significantly different from the age stratification of the high-risk population in the Global Burden of Disease study. It is further explained that the complex background of MHD patients (toxin accumulation, micro-inflammatory state, nutrient loss, uncontrolled blood pressure, prevalence of HF, etc.) makes the risk population of pneumonia in this population more younger (Ebert et al., 2020; Torres et al., 2021).

In this study, we included more variables related to HF management and dialysis treatment, and further evaluated their correlation with pneumonia. Our model demonstrated outstanding predictive performance, optimizing patient outcomes through the analysis of diverse metrics like ROC curves, calibration plots, and DCA curves. Nevertheless, there are still some limitations in our research. First of all, this study is a retrospective cohort study involving two centers. The clinical data of two hospitals located in different geographical regions were collected. Through the comparison of baseline data between groups, it can be found that there are significant differences between different centers. The differences in disease diagnosis and treatment strategies between different centers may directly affect the results and accuracy of the prediction model. Secondly, despite the external validation of this study, the sample size remains relatively small, potentially restricting the generalizability and statistical reliability of the findings. Future clinical studies are anticipated to utilize a larger sample size to provide additional validation for the model.

Conclusions

Our study established and validated a pneumonia prediction model specifically for patients undergoing MHD. The model developed by showed excellent predictive performance and was a valuable tool for assessing the risk of pneumonia in subgroup patients.

Supplemental Information

Supplemental Information 1 Baseline data of 405 patients

Supplemental Information 2 Codebook

Supplemental Information 3 Data density distribution map

Supplemental Information 4 Analysis of the relationship between screened predictors

Supplemental Information 5 Scaled Schoenfeld residual plots for predictors against time in the dataset

Supplemental Information 6 Survival probability of seven factors based on the Kaplan–Meier analysis

Supplemental Information 7 Abbreviations

Supplemental Information 8 Data defect interpolation

Supplemental Information 9 Single-factor Cox analysis

Thanks to Extreme Smart Analysis for technical support.

Additional Information and Declarations

Competing Interests

Author Contributions

Human Ethics

Clinical Trial Ethics

Data Availability

Clinical Trial Registration

The authors declare there are no competing interests.

Xiaohua Yang conceived and designed the experiments, performed the experiments, analyzed the data, prepared figures and/or tables, and approved the final draft.

Ju Zhang conceived and designed the experiments, performed the experiments, prepared figures and/or tables, and approved the final draft.

Xisheng Xie conceived and designed the experiments, authored or reviewed drafts of the article, and approved the final draft.

Wenwu Tang conceived and designed the experiments, prepared figures and/or tables, and approved the final draft.

The following information was supplied relating to ethical approvals (i.e., approving body and any reference numbers):

The Medical Ethics Committee of Guangyuan Central Hospital. Approval No: 2024-08 and date of approval 11.19.2024.

The following information was supplied relating to ethical approvals (i.e., approving body and any reference numbers):

The Medical Ethics Committee of Guangyuan Central Hospital.

The following information was supplied regarding data availability:

Data are available in the Supplemental Files.

The following information was supplied regarding Clinical Trial registration:

No: 2024-08 and date of approval 11.19.2024.

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
