# Peer review of "Development and external validation of a multivariate model for predicting pneumonia in patients receiving maintenance hemodialysis: a retrospective study"

_PeerJ, doi:10.7717/peerj.20070_

## Round 0.1 · original submission · Major Revisions

Please make it clear what the contribution of this work is in light of existing publications in this area.

·

Basic reporting

In this paper authors reported risk factors for pneumnoia development in chronic hemodialysis. patients. They found that seven factors such as age, diabetes, LVMI, pleural fluid, WBC, A/G, CHD.
This study is retrospective. exclusion and inclusion criterias can be acceptable. statistically analysis s enough and appropriate. Tables and figures are enough

Experimental design

This study retrosective . the nmber of patients are enough. criterias of exclusion/inclusion are good. the criterias of diagnosis of neumonia are acceptible
statically method is enough

Validity of the findings

The results are presented in tables and figures. These are descriptive and enough

Additional comments

All results were discussed with literature support.
This section can be shortened

Reviewer 2 ·

Basic reporting

The authors performed a backward-looking analysis of risk assessment and prediction models for pneumonia in Chinese maintenance hemodialysis patients. Although the analytical methods and data presentation are precise, modifications are necessary for publication in this journal. The modifications are described below.
Background of the patient population is not provided.
Epidemiological studies of hemodialysis patients must take into account the characteristics of the region. Indication of basic patient background is indispensable for this purpose. Failure to present information such as patient age, gender, underlying disease leading to renal failure, type of dialyzer, dialysis volume such as KT/V, dialysis time, and medications will compromise the reliability of the analysis results.

Experimental design

There is no problem with the analytical methods, but I feel that too many analytical methods are being used too often, and conversely, the reliability of the models presented is being compromised with respect to their validity.

Validity of the findings

The results presented are very valid, but the differences from the results of previous studies seem unclear.

Additional comments

None.

---

## Round 0.2 · Minor Revisions

Reviewer 2 has some additional comments for you to address.

·

Basic reporting

IT CAN BE ACCEPTABLE

Experimental design

GOOD

Validity of the findings

ENOUGH

Additional comments

EVERYTHING IS ACCEPTABL

Reviewer 2 ·

Basic reporting

The patient backgrounds that I pointed out in my initial peer review were discussed in detail. This seems to have increased the credibility of this study.

Experimental design

There is no problem.

Validity of the findings

The article shows good validity for publication.

Additional comments

While the effort to increase the credibility of the study is well done, the discussion part is redundant. The attempt to explain the mechanism of the pneumonia-related factors extracted in the model created is commendable, but it is felt to be a bit of an over-estimation. The discussion section should be reduced to about half of the current volume.

---

## Round 0.3 · accepted · Accept

The authors have addressed all of the reviewers' comments and the manuscript is ready for publication.

Reviewer 2 ·

Basic reporting

The authors appropriately revised this manuscript according to the suggestions.

Experimental design

-

Validity of the findings

The results are not significantly different from those previously reported, but they can be evaluated as indicating the characteristics of a particular region.